# Bandit Samplers for Training Graph Neural Networks

**Ziqi Liu**[*]
Ant Group
ziqiliu@antfin.com

**Zhengwei Wu**[*]
Ant Group
zejun.wzw@antfin.com

**Zhiqiang Zhang**
Ant Group
lingyao.zzq@antfin.com

**Jun Zhou**[†]
Ant Group
jun.zhoujun@antfin.com

**Shuang Yang**
Alibaba Group
shuang.yang@antfin.com

**Le Song**
Ant Group
Georgia Institute of Technology
lsong@cc.gatech.edu

**Yuan Qi**
Ant Group
yuan.qi@antfin.com

## Abstract

Several sampling algorithms with variance reduction have been proposed for accelerating the training of Graph Convolution Networks (GCNs). However, due to the intractable computation of optimal sampling distribution, these sampling algorithms are suboptimal for GCNs and are not applicable to more general graph neural networks (GNNs) where the message aggregator contains learned weights rather than fixed weights, such as Graph Attention Networks (GAT). The fundamental reason is that the embeddings of the neighbors or learned weights involved in the optimal sampling distribution are *changing* during the training and *not known a priori*, but only *partially observed* when sampled, thus making the derivation of an optimal variance reduced samplers non-trivial. In this paper, we formulate the optimization of the sampling variance as an adversary bandit problem, where the rewards are related to the node embeddings and learned weights, and can vary constantly. Thus a good sampler needs to acquire variance information about more neighbors (exploration) while at the same time optimizing the immediate sampling variance (exploit). We theoretically show that our algorithm asymptotically approaches the optimal variance within a factor of 3. We show the efficiency and effectiveness of our approach on multiple datasets.

## 1 Introduction

Graph neural networks [13, 11] have emerged as a powerful tool for representation learning of graph data in irregular or non-euclidean domains [3, 21]. For instance, graph neural networks have demonstrated state-of-the-art performance on learning tasks such as node classification, link and graph property prediction, with applications ranging from drug design [8], social networks [11], transaction networks [14], gene expression networks [9], and knowledge graphs [17].

One major challenge of training GNNs comes from the requirements of heavy floating point operations and large memory footprints, due to the recursive expansions over the neighborhoods. For a minibatch with a single vertex $v_i$, to compute its embedding $h_i^{(L)}$ at the $L$-th layer, we have to expand its

---

[*]Equal contribution.
[†]Corresponding author.

neighborhood from the $(L-1)$-th layer to the 0-th layer, i.e. $L$-hops neighbors. That will soon cover a large portion of the graph if particularly the graph is dense. One basic idea of alleviating such "neighbor explosion" problem was to sample neighbors in a top-down manner, i.e. sample neighbors in the $l$-th layer given the nodes in the $(l+1)$-th layer recursively.

Several layer sampling approaches [11, 6, 12, 23] have been proposed to alleviate above "neighbor explosion" problem and improve the convergence of training GCNs, e.g. with importance sampling. However, the optimal sampler [12], $q_{ij}^\star = \frac{\alpha_{ij}\|h_j^{(l)}\|^2}{\sum_{k\in\mathcal{N}_i}\alpha_{ik}\|h_k^{(l)}\|^2}$ for vertex $v_i$, to minimize the variance of the estimator $\hat{h}_i^{(l+1)}$ involves all its neighbors' hidden embeddings, i.e. $\{\hat{h}_j^{(l)}|v_j\in\mathcal{N}_i\}$, which is infeasible to be computed because we can only observe them partially while doing sampling. Existing approaches [6, 12, 23] typically compromise the optimal sampling distribution via approximations, which may impede the convergence. Moreover, such approaches are not applicable to more general cases where the weights or kernels $\alpha_{ij}$'s are not known a priori, but are learned weights parameterized by attention functions [20]. That is, both the hidden embeddings and learned weights involved in the optimal sampler constantly *vary* during the training process, and only *part* of the unnormalized attention values or hidden embeddings can be observed while do sampling.

**Present work**. We derive novel variance reduced samplers for training of GCNs and attentive GNNs with a fundamentally different perspective. That is, different with existing approaches that need to compute the immediate sampling distribution, we maintain nonparametric estimates of the sampler instead, and update the sampler towards optimal variance after we acquire partial knowledges about neighbors being sampled, as the algorithm iterates.

To fulfil this purpose, we formulate the optimization of the samplers as a bandit problem, where the regret is the gap between expected loss (negative reward) under current policy (sampler) and expected loss with optimal policy. We define the reward with respect to each action, i.e. the choice of a set of neighbors with sample size $k$, as the derivatives of the sampling variance, and show the variance of our samplers asymptotically approaches the optimal variance within a factor of 3. Under this problem formulation, we propose two bandit algorithms. The first algorithm based on multi-armed bandit (MAB) chooses $k < K$ arms (neighbors) repeatedly. Our second algorithm based on MAB with multiple plays chooses a combinatorial set of neighbors with size $k$ only once.

To summarize, (**1**) We recast the sampler for GNNs as a bandit problem from a fundamentally different perspective. It works for GCNs and attentive GNNs while existing approaches apply only to GCNs. (**2**) We theoretically show that the regret with respect to the variance of our estimators asymptotically approximates the optimal sampler within a factor of 3 while no existing approaches optimize the sampler. (**3**) We empirically show that our approachs are way competitive in terms of convergence and sample variance, compared with state-of-the-art approaches on multiple public datasets.

## 2  Problem Setting

Let $\mathcal{G} = (\mathcal{V}, \mathcal{E})$ denote the graph with $N$ nodes $v_i \in \mathcal{V}$, and edges $(v_i, v_j) \in \mathcal{E}$. Let the adjacency matrix denote as $A \in \mathbb{R}^{N\times N}$. Assuming the feature matrix $H^{(0)} \in \mathbb{R}^{N\times D^{(0)}}$ with $h_i^{(0)}$ denoting the $D^{(0)}$-dimensional feature of node $v_i$. We focus on the following simple but general form of GNNs:

$$h_i^{(l+1)} = \sigma\Big(\sum_{j=1}^{N}\alpha(v_i,v_j)\,h_j^{(l)}\,W^{(l)}\Big),\quad l=0,\ldots,L-1 \tag{1}$$

where $h_i^{(l)}$ is the hidden embedding of node $v_i$ at the $l$-th layer, $\boldsymbol{\alpha} = (\alpha(v_i,v_j)) \in \mathbb{R}^{N\times N}$ is a kernel or weight matrix, $W^{(l)} \in \mathbb{R}^{D^{(l)}\times D^{(l+1)}}$ is the transform parameter on the $l$-th layer, and $\sigma(\cdot)$ is the activation function. The weight $\alpha(v_i, v_j)$, or $\alpha_{ij}$ for simplicity, is non-zero only if $v_j$ is in the 1-hop neighborhood $\mathcal{N}_i$ of $v_i$. It varies with the aggregation functions [3, 21]. For example, (**1**) GCNs [8, 13] define fixed weights as $\boldsymbol{\alpha} = \tilde{D}^{-1}\tilde{A}$ or $\boldsymbol{\alpha} = \tilde{D}^{-\frac{1}{2}}\tilde{A}\tilde{D}^{-\frac{1}{2}}$ respectively, where $\tilde{A} = A+I$, and $\tilde{D}$ is the diagonal node degree matrix of $\tilde{A}$. (**2**) The attentive GNNs [20, 15] define a learned weight $\alpha(v_i, v_j)$ by attention functions: $\alpha(v_i, v_j) = \frac{\tilde{\alpha}(v_i,v_j;\theta)}{\sum_{v_k\in\mathcal{N}_i}\tilde{\alpha}(v_i,v_k;\theta)}$, where the unnormalized attentions $\tilde{\alpha}(v_i,v_j;\theta) = \exp(\mathrm{ReLU}(a^T[Wh_i\|Wh_j]))$, are parameterized by $\theta = \{a, W\}$. Different

from GCNs, the learned weights $\alpha_{ij} \propto \tilde{\alpha}_{ij}$ can be evaluated only given all the unnormalized weights in the neighborhood.

The basic idea of layer sampling approaches [11, 6, 12, 23] was to recast the evaluation of Eq. (1) as

$$\hat{h}_i^{(l+1)} = \sigma \left( N(i) \, \mathbb{E}_{p_{ij}} \left[ \hat{h}_j^{(l)} \right] W^{(l)} \right), \tag{2}$$

where $p_{ij} \propto \alpha_{ij}$, and $N(i) = \sum_j \alpha_{ij}$. Hence we can evaluate each node $v_i$ at the $(l+1)$-th layer, using a Monte Carlo estimator with sampled neighbors at the $l$-th layer. Without loss of generality, we assume $p_{ij} = \alpha_{ij}$ and $N(i) = 1$ that meet the setting of attentive GNNs in the rest of this paper. To further reduce the variance, let us consider the following importance sampling

$$\hat{h}_i^{(l+1)} = \sigma_{W^{(l)}} \left( \hat{\mu}_i^{(l)} \right) = \sigma_{W^{(l)}} \left( \mathbb{E}_{q_{ij}} \left[ \frac{\alpha_{ij}}{q_{ij}} \hat{h}_j^{(l)} \right] \right), \tag{3}$$

where we use $\sigma_{W^{(l)}}(\cdot)$ to include transform parameter $W^{(l)}$ into the function $\sigma(\cdot)$ for conciseness. As such, one can find an alternative sampling distribution $q_i = (q_{ij_1}, ..., q_{ij_{|\mathcal{N}_i|}})$ to reduce the variance of an estimator, e.g. a Monte Carlo estimator $\hat{\mu}_i^{(l)} = \frac{1}{k} \sum_{s=1}^{k} \frac{\alpha_{ij_s}}{q_{ij_s}} \hat{h}_{j_s}^{(l)}$, where $j_s \sim q_i$.

Take expectation over $q_i$, we define the variance of $\hat{\mu}_i^{(l)} = \frac{\alpha_{ij_s}}{q_{ij_s}} \hat{h}_{j_s}^{(l)}$ at step $t$ and $(l+1)$-th layer to be:

$$\mathbb{V}^t(q_i) = \mathbb{E} \left[ \left\| \hat{\mu}_i^{(l)}(t) - \mu_i^{(l)}(t) \right\|^2 \right] = \mathbb{E} \left[ \left\| \frac{\alpha_{ij_s}(t)}{q_{ij_s}} h_{j_s}^{(l)}(t) - \sum_{j \in \mathcal{N}_i} \alpha_{ij}(t) h_j^{(l)}(t) \right\|^2 \right]. \tag{4}$$

Note that $\alpha_{ij}$ and $h(v_j)$ that are inferred during training may vary over steps $t$'s. We will explicitly include step $t$ and layer $l$ only when it is necessary. By expanding Eq. (4) one can write $\mathbb{V}(q_i)$ as the difference of two terms. The first is a function of $q_i$, which we refer to as the *effective variance*:

$$\mathbb{V}_e(q_i) = \sum_{j \in \mathcal{N}_i} \frac{1}{q_{ij}} \alpha_{ij}^2 \left\| h_j \right\|^2, \tag{5}$$

while the second does not depend on $q_i$, and we denote it by $\mathbb{V}_c = \left\| \sum_{j \in \mathcal{N}_i} \alpha_{ij} h_j \right\|^2$. The optimal sampling distribution [6, 12] at $(l+1)$-th layer for vertex $i$ that minimizes the variance is:

$$q_{ij}^{\star} = \frac{\alpha_{ij} \| h_j^{(l)} \|^2}{\sum_{k \in \mathcal{N}_i} \alpha_{ik} \| h_k^{(l)} \|^2}. \tag{6}$$

However, evaluating this sampling distribution is infeasible because we cannot have all the knowledges of neighbors' embeddings in the denominator of Eq. (6). Moreover, the $\alpha_{ij}$'s in attentive GNNs could also vary during the training procedure. Existing layer sampling approaches based on importance sampling just ignore the effects of norm of embeddings and assume the $\alpha_{ij}$'s are fixed during training. As a result, the sampling distribution is suboptimal and only applicable to GCNs where the weights are fixed. Note that our derivation above follows the setting of node-wise sampling approaches [11], but the claim remains to hold for layer-wise sampling approaches [6, 12, 23].

## 3   Related Works

We summarize three types of works for training graph neural networks.

First, several "*layer sampling*" approaches [11, 6, 12, 23] have been proposed to alleviate the "neighbor explosion" problems. Given a minibatch of labeled vertices at each iteration, such approaches sample neighbors layer by layer in a top-down manner. Particularly, node-wise samplers [11] randomly sample neighbors in the lower layer given each node in the upper layer, while layer-wise samplers [6, 12, 23] leverage importance sampling to sample neighbors in the lower layer given all the nodes in upper layer with sample sizes of each layer be independent of each other. Empirically, the layer-wise samplers work even worse [5] compared with node-wise samplers, and one can set an appropriate sample size for each layer to alleviate the growth issue of node-wise samplers. In this paper, we focus on optimizing the variance in the vein of layer sampling approaches. Though the

derivation of our bandit samplers follows the node-wise samplers, it can be extended to layer-wise. We leave this extension as a future work.

Second, Chen et al. [5] proposed a variance reduced estimator by maintaining historical embeddings of each vertices, based on the assumption that the embeddings of a single vertex would be close to its history. This estimator uses a simple random sampler and works efficient in practice at the expense of requiring an extra storage that is linear with number of nodes.

Third, two "*graph sampling*" approaches [7, 22] first cut the graph into partitions [7] or sample into subgraphs [22], then they train models on those partitions or subgraphs in a batch mode [13]. They show that the training time of each epoch may be much faster compared with "layer sampling" approaches. We summarize the drawbacks as follows. First, the partition of the original graph could be sensitive to the training problem. Second, these approaches assume that all the vertices in the graph have labels, however, in practice only partial vertices may have labels [14].

**GNNs Architecture.** For readers who are interested in the works related to the architecture of GNNs, please refer to the comprehensive survey [21]. Existing sampling approaches works only on GCNs, but not on more advanced architectures like GAT [20].

## 4 Variance Reduced Samplers as Bandit Problems

We formulate the optimization of sampling variance as a bandit problem. Basically, optimal variance requires the knowledge of all the neighbors' embeddings that are computation infeasible, and our chance is to exploit the sampled good neighbors. Our basic idea is that instead of explicitly calculating the intractable optimal sampling distribution in Eq. (6) at each iteration, we aim to optimize a sampler or **policy** $Q_i^t$ for each vertex $i$ over the horizontal steps $1 \leq t \leq T$, and make the variance of the estimator following this sampler asymptotically approach the optimum $Q_i^\star = \underset{Q_i}{\operatorname{argmin}} \sum_{t=1}^T \mathbb{V}_e^t(Q_i)$,

such that $\sum_{t=1}^T \mathbb{V}_e^t(Q_i^t) \leq c \sum_{t=1}^T \mathbb{V}_e^t(Q_i^\star)$ for some constant $c > 1$. Each **action** of policy $Q_i^t$ is a choice of any $k$-element set of sampled neighbors $S_i \subset \mathcal{N}_i$ where $S_i \sim Q_i^t$. We denote $Q_{i,S_i}(t)$ as the probability of the action that $v_i$ chooses $S_i$ at $t$. The gap to be minimized between effective variance and the oracle is

$$\mathbb{V}_e^t(Q_i^t) - \mathbb{V}_e^t(Q_i^\star) \leq \langle Q_i^t - Q_i^\star, \nabla_{Q_i^t} \mathbb{V}_e^t(Q_i^t) \rangle. \tag{7}$$

Note that the function $\mathbb{V}_e^t(Q_i^t)$ is convex w.r.t $Q_i^t$, hence for $Q_i^t$ and $Q_i^\star$ we have the upper bound derived on right hand of Eq. (7). We define this upper bound as **regret** at $t$, which means the expected loss (negative reward) with policy $Q_i^t$ minus the expected loss with optimal policy $Q_i^\star$. Hence the **reward** w.r.t choosing $S_i$ at $t$ is the negative derivative of the effective variance $r_{i,S_i}(t) = -\nabla_{Q_{i,S_i}(t)} \mathbb{V}_e^t(Q_i^t)$.

In the following, we adapt this bandit problem in the adversary bandit setting [1] because the rewards vary as the training proceeds and do not follow a priori fixed distribution [4]. We leave the studies of other bandits as a future work. We show in section 6 that with this regret the variances of our estimators asymptotically approach the optimal variance within a factor of 3.

Following importance sampling, both of our samplers maintain the alternative sampling distribution $q_i^t = (q_{ij_1}(t), ..., q_{ij_{|\mathcal{N}_i|}}(t))$ for each vertex $v_i$ over steps $t$'s. We instantiate above framework under two bandit settings. **(1)** In the adversary MAB setting [1], we define the sampler $Q_i^t$ as $q_i^t$, that samples exact an **arm** (neighbor) $v_{j_s} \subset \mathcal{N}_i$ from $q_i^t$. In this case the set $S_i$ is the element $v_{j_s}$. To have a sample size of $k$ neighbors, we repeat this process $k$ times. After we collected $k$ rewards $r_{ij_s}(t) = -\nabla_{q_{i,j_s}(t)} \mathbb{V}_e^t(q_i^t)$ we update $q_i^t$ by **EXP3** [1]. **(2)** In the adversary MAB with multiple plays setting [19], it uses an efficient $k$-combination sampler (**DepRound** [10]) $Q$ to sample any $k$-element subset $S \subset \{1, 2, ..., K\}$ that satisfies $\sum_{S:j \in S} Q_S = q_j, \forall j \in \{1, 2, ..., K\}$, where $q_j$ corresponds to the alternative probability of sampling $j$. As such, it allows us to select a set of $k$ distinct **arms** (neighbors) $S = \binom{K}{k}$ from $K$ arms at once. The selection can be done in $O(K)$. After we collected the reward $-\nabla_{Q_{i,S_i}(t)} \mathbb{V}_e^t(Q_i^t)$, we update $q_i^t$ by **EXP3.M** [19].

**Discussions.** We have to select a sample size of $k$ neighbors in GNNs. Note that in MAB setting, exact one neighbor should be selected and followed by updating the policy. Hence strictly speaking applying MAB to our problem is not rigorous. Applying MAB with multiple plays to our problem is rigorous because it allows us to sample $k$ neighbors at once and update the rewards together.

# 5 Algorithms

The framework of our algorithm is: **(1)** pick $k$ arms with a *sampler based on the alternative sampling distribution* $q_i^t$ for any vertex $v_i$, **(2)** establish the *unbiased estimator*, **(3)** do feedforward and backpropagation, and finally **(4)** *calculate the rewards* and *update the sampler* with a proper bandit algorithm. We show this framework in Algorithm 1. Note that the variance w.r.t $q_i$ in Eq. (4) is defined only at the $(l+1)$-th layer, hence we should maintain multiple $q_i$'s at each layer. In practice, we find that maintain a single $q_i$ and update it only using rewards from the 1-st layer works well enough. The time complexity of our algorithm is same with any node-wise approaches [11]. In addition, it requires a storage in $O(|\mathcal{E}|)$ to maintain the alternative sampling distribution, $|\mathcal{E}|$ is the number of edges used for message passing operations in GNNs. Beyond that, no further storage is required. This is true even for more sophisticated architectures where messages are passed between neighbors beyond one hop.

It remains to instantiate the estimators, variances and rewards related to our two bandit settings. We name our first algorithm **GNN-BS** under adversary MAB setting, and the second **GNN-BS.M** under adversary MAB with multiple plays setting. We first assume the weights $\alpha_{ij}$'s are fixed, then extend to attentive GNNs that $\alpha_{ij}(t)$'s change.

---

**Algorithm 1** Bandit Samplers for Training GNNs.

---

**Require:** step $T$, sample size $k$, number of layers $L$, node features $H^{(0)}$, adjacency matrix $A$.
1: **Initialize:** $q_{ij}(1) = 1/|\mathcal{N}_i|$ if $j \in \mathcal{N}_i$ else 0, $w_{ij}(1) = 1$ if $j \in \mathcal{N}_i$ else 0.
2: **for** $t = 1$ to $T$ **do**
3:     Read a minibatch of labeled vertices at layer $L$.
4:     Use sampler $q_i^t$ or **DepRound**$(k, q_i^t)$ to sample neighbors top-down with sample size $k$.
5:     Forward GNN model via estimators defined in Eq. (8) or Proposition 1.
6:     Backpropagation and update GNN model.
7:     **for** each $v_i$ in the 1-st layer **do**
8:         Collect $v_i$'s $k$ sampled neighbors $v_j \in S_i^t$, and rewards $r_i^t = \{r_{ij}(t) : v_j \in S_i^t\}$.
9:         Update $q_i^{t+1}$ and $w_i^{t+1}$ by **EXP3**$(q_i^t, w_i^t, r_i^t, S_i^t)$ or **EXP3.M**$(q_i^t, w_i^t, r_i^t, S_i^t)$.
10:     **end for**
11: **end for**
12: **return** GNN model.

---

## 5.1 GNN-BS: Graph Neural Networks with Bandit Sampler

In this setting, we choose 1 arm and repeat $k$ times. We have the following Monte Carlo estimator

$$\hat{\mu}_i = \frac{1}{k} \sum_{s=1}^k \frac{\alpha_{ij_s}}{q_{ij_s}} \hat{h}_{j_s}, \ j_s \sim q_i. \tag{8}$$

This yields the variance $\mathbb{V}(q_i) = \frac{1}{k} \mathbb{E}_{q_i} \left[ \left\| \frac{\alpha_{ij_s}}{q_{ij_s}} h_{j_s} - \sum_{j \in \mathcal{N}_i} \alpha_{ij} h_j \right\|^2 \right]$. Following Eq. (5) and Eq. (7), we have the reward of $v_i$ picking neighbor $v_j$ at step $t$ as

$$r_{ij}(t) = -\nabla_{q_{ij}(t)} \mathbb{V}_e^t(q_i^t) = \frac{\alpha_{ij}^2}{k \cdot q_{ij}(t)^2} \|h_j(t)\|^2. \tag{9}$$

## 5.2 GNN-BS.M: Graph Neural Networks with Multiple Plays Bandit Sampler

Given a vertex $v_i$, an important property of DepRound is that it satisfies $\sum_{S_i:j \in S_i} Q_{i,S_i} = q_{ij}, \forall v_j \in \mathcal{N}_i$, where $S_i \subset \mathcal{N}_i$ is any subset of size $k$. We have the following unbiased estimator.

**Proposition 1.** $\hat{\mu}_i = \sum_{j_s \in S_i} \frac{\alpha_{ij_s}}{q_{ij_s}} h_{j_s}$ *is the unbiased estimator of* $\mu_i = \sum_{j \in \mathcal{N}_i} \alpha_{ij} h_j$ *given that* $S_i$ *is sampled from* $Q_i$ *with DepRound, where* $S_i$ *is the selected $k$-subset neighbors of vertex* $i$.

The effective variance of this estimator is $\mathbb{V}_e(Q_i) = \sum_{S_i \subset \mathcal{N}_i} Q_{i,S_i} \| \sum_{j_s \in S_i} \frac{\alpha_{ij_s}}{q_{ij_s}} h_{j_s} \|^2$. Since the derivative of this effective variance w.r.t $Q_{i,S_i}$ does not factorize, we instead have the following approximated effective variance using Jensen's inequality.

**Proposition 2.** *The effective variance can be approximated by* $\mathbb{V}_e(Q_i) \leq \sum_{j_s \in \mathcal{N}_i} \frac{\alpha_{ij_s}}{q_{ij_s}} \|h_{j_s}\|^2$.

**Proposition 3.** *The negative derivative of the approximated effective variance* $\sum_{j_s \in \mathcal{N}_i} \frac{\alpha_{ij_s}}{q_{ij_s}} \|h_{j_s}\|^2$ *w.r.t* $Q_{i,S_i}$, *i.e. the reward of* $v_i$ *choosing* $S_i$ *at* $t$ *is* $r_{i,S_i}(t) = \sum_{j_s \in S_i} \frac{\alpha_{ij_s}}{q_{ij_s}(t)^2} \|h_{j_s}(t)\|^2$.

Follow EXP3.M we use the reward w.r.t each arm as $r_{ij}(t) = \frac{\alpha_{ij}}{q_{ij}(t)^2} \|h_j(t)\|^2, \forall j \in S_i$. Our proofs rely on the property of DepRound introduced above.

### 5.3 Extension to Attentive GNNs

In this section, we extend our algorithms to attentive GNNs. The issue remained is that the attention value $\alpha_{ij}$ can not be evaluated with only sampled neighborhoods, instead, we can only compute the unnormalized attentions $\tilde{\alpha}_{ij}$. We define the adjusted feedback attention values as follows:

$$\alpha'_{ij} = \sum_{j \in S_i} q_{ij} \cdot \frac{\tilde{\alpha}_{ij}}{\sum_{j \in S_i} \tilde{\alpha}_{ij}}, \tag{10}$$

where $\tilde{\alpha}_{ij}$'s are the unnormalized attention values that can be obviously evaluated when we have sampled $(v_i, v_j)$. We use $\sum_{j \in S_i} q_{ij}$ as a surrogate of $\frac{\sum_{j \in S_i} \tilde{\alpha}_{ij}}{\sum_{j \in \mathcal{N}_i} \tilde{\alpha}_{ij}}$ so that we can approximate the truth attention values $\alpha_{ij}$ by our adjusted attention values $\alpha'_{ij}$.

## 6 Regret Analysis

As we described in section 4, the regret is defined as $\langle Q_i^t - Q_i^\star, \nabla_{Q_i^t} \mathbb{V}_e^t(Q_i^t) \rangle$. By choosing the reward as the negative derivative of the effective variance, we have the following theorem that our bandit sampling algorithms asymptotically approximate the optimal variance within a factor of 3.

**Theorem 1.** *Using Algorithm 1 with* $\eta = 0.4$ *and* $\delta = \sqrt{\frac{(1-\eta)\eta^4 k^5 \ln(n/k)}{Tn^4}}$ *to minimize the effective variance with respect to* $\{Q_i^t\}_{1 \leq t \leq T}$, *we have*

$$\sum_{t=1}^{T} \mathbb{V}_e^t(Q_i^t) \leq 3 \sum_{t=1}^{T} \mathbb{V}_e^t(Q_i^\star) + 10\sqrt{\frac{Tn^4 \ln(n/k)}{k^3}} \tag{11}$$

*where* $T \geq \ln(n/k)n^2(1-\eta)/(k\eta^2)$, $n = |\mathcal{N}_i|$.

Our proof follows [16] by upper and lower bounding the potential function. The upper and lower bounds are the functions of the alternative sampling probability $q_{ij}(t)$ and the reward $r_{ij}(t)$ respectively. By multiplying the upper and lower bounds by the optimal sampling probability $q_i^\star$ and using the reward definition in (9), we have the upper bound of the effective variance. The growth of this regret is sublinear in terms of $T$. The regret decreases in polynomial as sample size $k$ grows. Note that the number of neighbors $n$ is always well bounded in pratical graphs, and can be considered as a moderate constant number. Compared with existing layer sampling approaches, this is the first work optimizing the sampling variance of GNNs towards optimum. We will empirically show the sampling variances in experiments.

## 7 Experiments

In this section, we conduct extensive experiments compared with state-of-the-art approaches to show the advantage of our training approaches. We use the following rule to name our approaches: GNN architecture plus bandit sampler. For example, **GCN-BS**, **GAT-BS** and **GP-BS** denote the training approaches for GCN, GAT [20] and GeniePath [15] respectively. Please find our implementations at https://github.com/xavierzw/gnn-bs. We run all the experiments using one machine with Intel Xeon E5-2682 and 512GB RAM.

The major purpose of this paper is to compare the effects of our samplers with existing training algorithms, so we compare them by training the same GNN architecture. We use the following architectures unless otherwise stated. We fix the number of layers as 2 as in [13] for all comparison

Table 1: Dataset summary. "s" dontes multi-class task, and "m" denotes multi-label task.

| Dateset | V | E | Degree | # Classes | # Features | # train | # validation | # test |
|---|---|---|---|---|---|---|---|---|
| Cora | $2,708$ | $5,429$ | 2 | 7 (s) | $1,433$ | $1,208$ | 500 | $1,000$ |
| Pubmed | $19,717$ | $44,338$ | 3 | 3 (s) | 500 | $18,217$ | 500 | $1,000$ |
| PPI | $56,944$ | $818,716$ | 15 | 121 (m) | 50 | $44,906$ | $6,514$ | $5,524$ |
| Reddit | $232,965$ | $11,606,919$ | 50 | 41 (s) | 602 | $153,932$ | $23,699$ | $55,334$ |
| Flickr | $89,250$ | $899,756$ | 10 | 7 (s) | 500 | $44,625$ | $22,312$ | $22,313$ |

algorithms. We set the dimension of hidden embeddings as 16 for Cora and Pubmed, and 256 for PPI, Reddit and Flickr. For a fair comparison, we do not use the normalization layer [2] particularly used in some works [5, 22]. For attentive GNNs, we use the attention layer proposed in GAT. we set the number of multi-heads as 1 for simplicity.

We report results on 5 benchmark data that include *Cora* [18], *Pubmed* [18], *PPI* [11], *Reddit* [11], and *Flickr* [22]. We follow the standard data splits, and summarize the statistics in Table 1.

We summarize the comparison algorithms as follows. **(1)** GraphSAGE [11] is a node-wise layer sampling approach with a random sampler. **(2)** FastGCN [6], LADIES [23], and AS-GCN [12] are layer sampling approaches based on importance sampling. **(3)** S-GCN [5] can be viewed as an optimization solver for training of GCN based on a simply random sampler. **(4)** ClusterGCN [7] and GraphSAINT [22] are "graph sampling" techniques that first partition or sample the graph into small subgraphs, then train each subgraph using the batch algorithm [13]. **(5)** The open source algorithms that support the training of attentive GNNs are AS-GCN and GraphSAINT. We denote them as AS-GAT and GraphSAINT-GAT.

We save the model based on the best results on validation and report results on testing data in Section 7.1. We do grid search for the following hyperparameters in each algorithm, i.e., the learning rate $\{0.01, 0.001\}$, the penalty weight on the $\ell_2$-norm regularizers $\{0, 0.0001, 0.0005, 0.001\}$, the dropout rate $\{0, 0.1, 0.2, 0.3\}$. By following the exsiting implementations[3], we save the model based on the best results on validation, and restore the model to report results on testing data in Section 7.1. For the sample size in GraphSAGE, S-GCN and our algorithms, we set 1 for Cora and Pubmed, 5 for Flickr, 10 for PPI and reddit. We set the sample size in the first and second layer for FastGCN/LADIES and AS-GCN/AS-GAT as 256 and 256 for Cora and Pubmed, $1,900$ and $3,800$ for PPI, 780 and $1,560$ for Flickr, and $2,350$ and $4,700$ for Reddit. We set the batch size of all the layer sampling approaches and S-GCN as 256 for all the datasets. For ClusterGCN, we set the partitions according to the suggestions [7] for PPI and Reddit. We set the number of partitions for Cora and Pubmed as 10, for flickr as 200 by doing grid search. We set the architecture of GraphSAINT as "0-1-1"[4] which means MLP layer followed by two graph convolution layers. We use the "rw" sampling strategy that reported as the best in their original paper to perform the graph sampling procedure. We set the number of root and walk length as the paper suggested.

Table 2: Comparisons on the GCN architecture: testing Micro F1 scores.

| Method | Cora | Pubmed | PPI | Reddit | Flickr |
|---|---|---|---|---|---|
| GraphSAGE | $0.731(\pm0.014)$ | $0.890(\pm0.002)$ | $0.689(\pm0.005)$ | $0.949(\pm0.001)$ | $0.494(\pm0.001)$ |
| FastGCN | $0.827(\pm0.001)$ | $0.895(\pm0.005)$ | $0.502(\pm0.003)$ | $0.825(\pm0.006)$ | $0.500(\pm0.001)$ |
| LADIES | $0.843(\pm0.003)$ | $0.880(\pm0.006)$ | $0.574(\pm0.003)$ | $0.932(\pm0.001)$ | $0.465(\pm0.007)$ |
| AS-GCN | $0.830(\pm0.001)$ | $0.888(\pm0.006)$ | $0.599(\pm0.004)$ | $0.890(\pm0.013)$ | $0.506(\pm0.012)$ |
| S-GCN | $0.828(\pm0.001)$ | $0.893(\pm0.001)$ | $0.744(\pm0.003)$ | $0.943(\pm0.001)$ | $0.501(\pm0.002)$ |
| ClusterGCN | $0.807(\pm0.006)$ | $0.887(\pm0.001)$ | $0.853(\pm0.001)$ | $0.938(\pm0.002)$ | $0.418(\pm0.002)$ |
| GraphSAINT | $0.815(\pm0.012)$ | $0.899(\pm0.002)$ | $0.787(\pm0.003)$ | $\mathbf{0.965}(\pm0.001)$ | $0.507(\pm0.001)$ |
| GCN-BS | $\mathbf{0.855}(\pm0.005)$ | $\mathbf{0.903}(\pm0.001)$ | $\mathbf{0.905}(\pm0.003)$ | $0.957(\pm0.000)$ | $\mathbf{0.513}(\pm0.001)$ |

Table 3: Comparisons on the attentive GNNs architecture: testing Micro F1 scores.

| Method | Cora | Pubmed | PPI | Reddit | Flickr |
|---|---|---|---|---|---|
| AS-GAT | 0.813(±0.001) | 0.884(±0.003) | 0.566(±0.002) | NA | 0.472(±0.012) |
| GraphSAINT-GAT | 0.773(±0.036) | 0.886(±0.016) | 0.789(±0.001) | 0.933(±0.012) | 0.470(±0.002) |
| GAT-BS | **0.857**(±0.003) | **0.894**(±0.001) | 0.841(±0.001) | 0.962(±0.001) | **0.513**(±0.001) |
| GAT-BS.M | **0.857**(±0.003) | **0.894**(±0.000) | 0.867(±0.003) | 0.962(±0.000) | **0.513**(±0.001) |
| GP-BS | 0.811(±0.002) | 0.890(±0.003) | 0.958(±0.001) | **0.964**(±0.000) | 0.507(±0.000) |
| GP-BS.M | 0.811(±0.001) | 0.892(±0.001) | **0.965**(±0.001) | **0.964**(±0.000) | 0.507(±0.000) |

## 7.1 Results on Benchmark Data

We report the testing results on GCN and attentive GNN architectures in Table 2 and Table 3 respectively. We run the results of each algorithm 3 times and report the mean and standard deviation. The results on the two layer GCN architecture show that our GCN-BS performs the best on most of datasets. Compared with layer sampling approaches, GCN-BS performs significantly better in relative dense graphs, such as PPI and Reddit. This shows the efficiency of our sampler on selecting neighbors. The results on the two layer attentive GNN architecture show the superiority of our algorithms on training more complex GNN architectures. GraphSAINT or AS-GAT do not compute the softmax of learned weights, but simply use the unnormalized weights to perform the aggregation. As a result, most of results from AS-GAT and GraphSAINT-GAT in Table 3 are worse than their results in Table 2. Thanks to the power of attentive structures in GNNs, our algorithms perform the best results on PPI and Flickr.

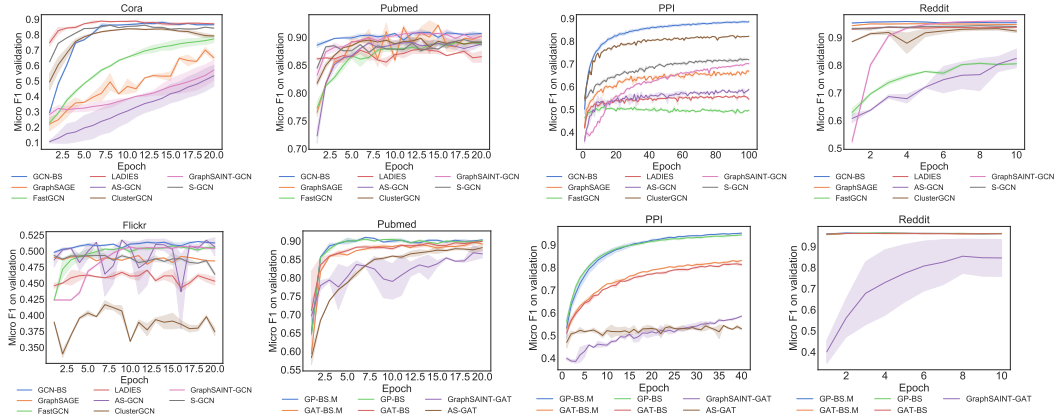

Figure 1: The convergence on validation in terms of epochs.

## 7.2 Convergence

In this section, we analyze the convergences of comparison algorithms on the two layer GCN and attentive GNN architectures in Figure 1 in terms of epoch. We run all the algorithms 3 times and show the mean and standard deviation. Our approaches converge much faster with lower variances in most datasets. The GNN-BS algorithms perform very similar to GNN-BS.M, even though strictly speaking GNN-BS does not follow the rigorous MAB setting.

The convergences on validation in terms of timing (seconds), compared with layer sampling approaches, in Fig. 2 show the similar results.

## 7.3 Sample Size Analysis

We analyze the sampling variances and accuracy as sample size varies using PPI data. Note that existing layer sampling approaches do not optimize the variances once the samplers are specified.

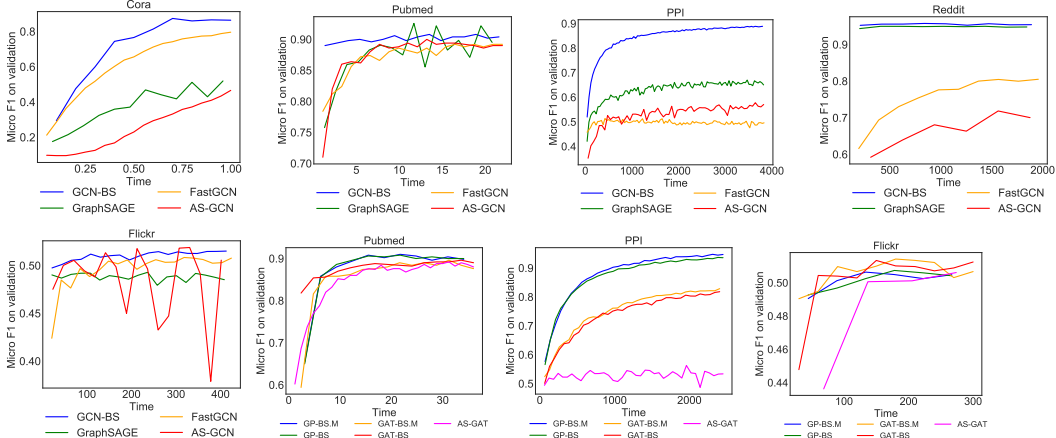

Figure 2: The convergence on validation in terms of timing.

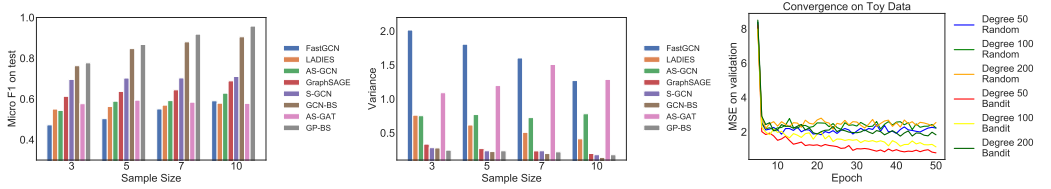

Figure 3: Comparisons on PPI by varying the sample sizes: F1 score (**left**), sample variances (**middle**). Convergence on toy data with different degrees (**right**).

As a result, their variances are simply fixed [23], while our approaches asymptotically appoach the optimum. For comparison, we train our models until convergence, then compute the average sampling variances. We show the results in Figure 3 (left and middle). The results are grouped in two categories, i.e. results for GCN and attentive GNNs respectively. Our approaches' sampling variances are smaller in each group. This explains the performances of our approaches on Micro F1 scores. Note that the overall sampling variances of node-wise approaches are way better than those of layer-wise approaches.

To further show the convergence while we simulate graphs with different degrees and fix the sample size of different algorithms, we set up the following experiments. We randomly sample 100 labeled nodes $\{1,...,i,...,100\}$ with each $\mu_i$ uniformly sampled from [-10, 10]. For each labeled node $i$ we generate $k$ neighbors, and its neighbors' features are 1-dimensional scalars in real field that are sampled from $\text{uniform}(\mu_i - \sigma, \mu_i + \sigma)$ with $\sigma = 5$. Each node $i$'s label is generated by simply averaging its neighbors' 1-dimensional scalar features. We use a GCN architecture with mean aggregators. We compare the convergence (mean squared error loss) with a random sampler by increasing $k = 50$ to 100 and 200 in Figure 3 (right). All the samplers use a fixed sample size 10. It shows that our bandit sampler works much better compared with a uniform sampler on graphs with different degrees.

## 8 Conclusions

In this paper, we show that the optimal layer samplers based on importance sampling for training general graph neural networks are computationally intractable, since it needs all the neighbors' hidden embeddings or learned weights. Instead, we re-formulate the sampling problem as a bandit problem that requires only partial knowledges from neighbors being sampled. We propose two algorithms based on multi-armed bandit and MAB with multiple plays. We show the variance of our bandit sampler asymptotically approaches the optimum within a factor of 3. We empirically show that our algorithms achieve much better convergence results with much lower variances compared with state-of-the-art approaches.

## Broader Impact

This paper presents an approach for fast training of graph neural networks with theoretical guarantees. It may have impacts on training approaches related to any models based on message passing. The graph neural networks may have positive impacts on recommendater systems, protein analyses, fraud detection and so on. This work does not present any foreseeable societal consequence.

## Acknowledgments and Disclosure of Funding

This work is supported by Ant Group.

## Footnotes

[3]Checkout: `https://github.com/matenure/FastGCN` or `https://github.com/huangwb/AS-GCN`

[4]Checkout `https://github.com/GraphSAINT/` for more details.

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
