[Supplementary Material]

# Supplementary materials for Paper "Bandit Samplers for Training Graph Neural Networks"

**Ziqi Liu**[*]
Ant Group
ziqiliu@antfin.com

**Zhengwei Wu**[*]
Ant Group
zejun.wzw@antfin.com

**Zhiqiang Zhang**
Ant Group
lingyao.zzq@antfin.com

**Jun Zhou**[†]
Ant Group
jun.zhoujun@antfin.com

**Shuang Yang**
Alibaba Group
shuang.yang@antfin.com

**Le Song**
Ant Group
Georgia Institute of Technology
lsong@cc.gatech.edu

**Yuan Qi**
Ant Group
yuan.qi@antfin.com

## A   Experiments

Figure 1: The convergence in timing (seconds) on GCNs.

[*]Equal Contribution.
[†]Corresponding author.

Figure 2: The convergence in timing (seconds) on attentive GNNs.

## A.1 Convergences

We show the convergences on validation in terms of timing (seconds) in Figure 1 and Figure 2. Basically, our algorithms converge to much better results in nearly same duration compared with other "layer sampling" approaches.

Note that we cannot complete the training of AS-GAT on Reddit because of memory issues.

## A.2 Discussions on Timings between Layer Sampling and Graph Sampling Paradigms

Note that the comparisons of timing between "graph sampling" and "layer sampling" paradigms have been studied recently in [1, 3]. As a result, we do not compare the timing with "graph sampling" approaches. Under certain conditions, the graph sampling approaches should be faster than layer sampling approaches. That is, graph sampling approaches are designed for graph data that all vertices have labels. Under such condition, the floating point operations analyzed in [1] are maximally utilized compared with the "layer sampling" paradigm. However, in practice, there are large amount of graph data with labels only on some types of vertices, such as the graphs in [2]. "Graph sampling" approaches are not applicable to cases where only partial vertices have labels. To summarize, the "layer sampling" approaches are more flexible and general compared with "graph sampling" approaches in many cases.

# B Algorithms

---

**Algorithm 2** EXP3($q_i^t, w_i^t, r_i^t, S_i^t$).

---

**Require:** $\eta = 0.4$, sample size $k$, neighbor size $n = |\mathcal{N}_i|$, $\delta = \sqrt{(1-\eta)\eta^4 k^5 \ln(n/k)/(Tn^4)}$.
  1: Set
$$\hat{r}_{ij}(t) = r_{ij}(t)/q_{ij}(t) \text{ if } j \in S_i^t \text{ else } 0$$
$$w_{ij}(t+1) = w_{ij}(t) \exp(\delta \, \hat{r}_{ij}(t)/n)$$
  2: Set $q_{ij}(t+1) \leftarrow (1-\eta)\frac{w_{ij}(t+1)}{\sum_{j \in \mathcal{N}_i} w_{ij}(t+1)} + \frac{\eta}{n}, \quad \text{for } j \in \mathcal{N}_i$

---

---
**Algorithm 3** DepRound$(k, (q_1, q_2, ..., q_K))$
---
1: **Input:** Sample size $k(k < K)$, sample distribution $(q_1, q_2, ..., q_K)$ with $\sum_{i=1}^K q_i = k$
2: **Output:** Subset of $[K]$ with $k$ elements
3: **while** there is an $i$ with $0 < q_i < 1$ **do**
4:     Choose distinct $i$ and $j$ with $0 < q_i < 1$ and $0 < q_j < 1$
5:     Set $\beta = \min\{1 - q_i, q_j\}$ and $\gamma = \min\{q_i, 1 - q_j\}$
6:     Update $q_i$ and $q_j$ as

$$(q_i, q_j) = \begin{cases} (q_i + \beta, q_j - \beta) \text{ with probability } \frac{\gamma}{\beta+\gamma} \\ (q_i - \gamma, q_j + \gamma) \text{ with probability } \frac{\beta}{\beta+\gamma} \end{cases}$$

7: **end while**
8: **return** $\{i : q_i = 1, 1 \le i \le K\}$

---

---
**Algorithm 4** EXP3.M$(q_i^t, w_i^t, r_i^t, S_i^t)$
---
**Require:** $\eta = 0.4$, sample size $k$, neighbor size $n = |\mathcal{N}_i|$, $\delta = \sqrt{(1 - \eta)\eta^4 k^5 \ln(n/k)/(Tn^4)}$, $U_i^t = \emptyset$.
1: For $j \in \mathcal{N}_i$ set

$$\hat{r}_{ij}(t) = \begin{cases} r_{ij}(t)/q_{ij}(t) & \text{if } j \in S_i^t \\ 0 & \text{otherwise} \end{cases}$$

$$w_{ij}(t + 1) = \begin{cases} w_{ij}(t) \exp(\delta \hat{r}_{ij}(t)/n) & \text{if } j \notin U_i^t \\ w_{ij}(t) & \text{otherwise} \end{cases}$$

2: **if** $\max_{j \in \mathcal{N}_i} w_{ij}(t + 1) \ge (\frac{1}{k} - \frac{\eta}{n}) \sum_{j \in \mathcal{N}_i} w_{ij}(t + 1)/(1 - \eta)$ **then**
3:     Decide $a_t$ so as to satisfy

$$\frac{a_t}{\sum_{w_{ij}(t+1) \ge a_t} a_t + \sum_{w_{ij}(t+1) < a_t} w_{ij}(t + 1)} = (\frac{1}{k} - \frac{\eta}{n})/(1 - \eta)$$

4:     Set $U_i^{t+1} = \{j : w_{ij}(t + 1) \ge a_t\}$
5: **else**
6:     Set $U_i^{t+1} = \emptyset$
7: **end if**
8: Set $w'_{ij}(t + 1) = \begin{cases} w_{ij}(t + 1) & \text{if } j \in \mathcal{N}_i \backslash U_i^{t+1} \\ a_t & \text{if } j \in U_i^t \end{cases}$
9: Set $q_{ij}(t + 1) = k\left((1 - \eta)\frac{w'_{ij}(t+1)}{\sum_{j \in \mathcal{N}_i} w'_{ij}(t+1)} + \frac{\eta}{n}\right)$ for $j \in \mathcal{N}_i$

---

## C Proofs

**Proposition 1.** $\hat{\mu}_i = \sum_{j_s \in S_i} \frac{\alpha_{ij_s}}{q_{ij_s}} h_{j_s}$ *is the unbiased estimator of* $\mu_i = \sum_{j \in \mathcal{N}_i} \alpha_{ij} h_j$ *given that* $S_i$ *is sampled from* $q_i$ *using the DepRound sampler* $Q_i$*, where* $S_i$ *is the selected* $k$*-subset neighbors of vertex* $i$.

*Proof.* Let us denote $Q_{i,S_i}$ as the probability of vertex $v_i$ choosing any $k$-element subset $S_i \subset \mathcal{N}_i$ from the $K$-element set $\mathcal{N}_i$ using DepRound sampler $Q_i$. This sampler follows the alternative sampling distribution $q_i = (q_{ij_1}, ..., q_{ij_K})$ where $q_{ij_s}$ denotes the alternative probability of sampling neighbor $v_{j_s}$. This sampler is guaranteed to satisfy $\sum_{S_i:j \in S_i} Q_{i,S_i} = q_{ij}$, i.e. the sum over the probabilities of all subsets $S_i$ that contains element $j$ equals the probability $q_{ij}$.

$$\mathbb{E}\left[\hat{\mu}_i\right] = \mathbb{E}\left[\sum_{j_s \in S_i} \frac{\alpha_{ij_s}}{q_{ij_s}} h_{j_s}\right] \tag{1}$$

$$= \sum_{S_i \subset \mathcal{N}_i} Q_{i,S_i} \sum_{j_s \in S_i} \frac{\alpha_{ij_s}}{q_{ij_s}} h_{j_s} \tag{2}$$

$$= \sum_{j \in \mathcal{N}_i} \sum_{S_i:j \in S_i} Q_{i,S_i} \frac{\alpha_{ij}}{q_{ij}} h_j \tag{3}$$

$$= \sum_{j \in \mathcal{N}_i} \frac{\alpha_{ij}}{q_{ij}} h_j \sum_{S_i:j \in S_i} Q_{i,S_i} \tag{4}$$

$$= \sum_{j \in \mathcal{N}_i} \frac{\alpha_{ij}}{q_{ij}} h_j q_{ij} \tag{5}$$

$$= \sum_{j \in \mathcal{N}_i} \alpha_{ij} h_j \tag{6}$$

$\square$

**Proposition 2.** *The effective variance can be approximated by* $\mathbb{V}_e(Q_i) \leq \sum_{j_s \in \mathcal{N}_i} \frac{\alpha_{ij_s}}{q_{ij_s}} \|h_{j_s}\|^2$.

*Proof.* The variance is

$$\mathbb{V}(Q_i) = \mathbb{E}\left[\left\|\sum_{j_s \in S_i} \frac{\alpha_{ij_s}}{q_{ij_s}} h_{j_s} - \sum_{j \in \mathcal{N}_i} \alpha_{ij} h_j\right\|^2\right]$$

$$= \sum_{S_i \subset \mathcal{N}_i} Q_{i,S_i} \left\|\sum_{j_s \in S_i} \frac{\alpha_{ij_s}}{q_{ij_s}} h_{j_s}\right\|^2 - \left\|\sum_{j \in \mathcal{N}_i} \alpha_{ij} h_j\right\|^2.$$

Therefore the effective variance has following upper bound:

$$\mathbb{V}_e(Q_i) = \sum_{S_i \subset \mathcal{N}_i} Q_{i,S_i} \left\|\sum_{j_s \in S_i} \frac{\alpha_{ij_s}}{q_{ij_s}} h_{j_s}\right\|^2$$

$$\leq \sum_{S_i \subset \mathcal{N}_i} Q_{i,S_i} \sum_{j_s \in S_i} \alpha_{ij_s} \left\|\frac{h_{j_s}}{q_{ij_s}}\right\|^2 \quad (Jensen's\ Inequality)$$

$$= \sum_{j_s \in \mathcal{N}_i} \sum_{S_i:j_s \in S_i} Q_{i,S_i} \alpha_{ij_s} \left\|\frac{h_{j_s}}{q_{ij_s}}\right\|^2$$

$$= \sum_{j_s \in \mathcal{N}_i} \frac{\alpha_{ij_s}}{q_{ij_s}^2} \|h_{j_s}\|^2 \sum_{S_i:j_s \in S_i} Q_{i,S_i}$$

$$= \sum_{j_s \in \mathcal{N}_i} \frac{\alpha_{ij_s}}{q_{ij_s}} \|h_{j_s}\|^2$$

□

**Proposition 3.** *The negative derivative of the approximated effective variance* $\sum_{j_s \in \mathcal{N}_i} \frac{\alpha_{ij_s}}{q_{ij_s}} \|h_{j_s}\|^2$
*w.r.t* $Q_{i,S_i}$*, i.e. the reward of* $v_i$ *choosing* $S_i$ *at* $t$*, is* $r_{i,S_i}(t) = \sum_{j_s \in S_i} \frac{\alpha_{ij_s}}{q_{ij_s}(t)^2} \|h_{j_s}(t)\|^2$.

*Proof.* Define the upper bound as $\hat{\mathbb{V}}_e(Q_i) = \sum_{j_s \in \mathcal{N}_i} \frac{\alpha_{ij_s}}{q_{ij_s}} \|h_{j_s}\|^2$, then its derivative is

$$
\begin{aligned}
\nabla_{Q_{i,S_i}} \hat{\mathbb{V}}_e(Q_i) &= \nabla_{Q_{i,S_i}} \sum_{j_s \in \mathcal{N}_i} \frac{\alpha_{ij_s}}{q_{ij_s}} \|h_{j_s}\|^2 \\
&= \nabla_{Q_{i,S_i}} \sum_{j_s \in \mathcal{N}_i} \frac{\alpha_{ij_s}}{\sum_{S'_i : j_s \in S'_i} Q_{i,S'_i}} \|h_{j_s}\|^2 \\
&= \nabla_{Q_{i,S_i}} \sum_{j_s \in S_i} \frac{\alpha_{ij_s}}{\sum_{S'_i : j_s \in S'_i} Q_{i,S'_i}} \|h_{j_s}\|^2 \\
&= - \sum_{j_s \in S_i} \frac{\alpha_{j_s}}{q_{ij_s}^2} \|h_{j_s}\|^2 \ (chain\, rule)
\end{aligned}
$$

□

Before we give the proof of Theorem 1, we first prove the following Lemma 1 that will be used later.

**Lemma 1.** *For any real value constant* $\eta \leq 1$ *and any valid distributions* $Q_i^t$ *and* $Q_i^\star$ *we have*

$$
(1 - 2\eta)\mathbb{V}_e^t(Q_i^t) - (1 - \eta)\mathbb{V}_e^t(Q_i^\star) \leq \langle Q_i^t - Q_i^\star, \nabla_{Q_i^t} \mathbb{V}_e^t(Q_i^t) \rangle + \eta \langle Q_i^\star, \nabla_{Q_i^t} \mathbb{V}_e^t(Q_i^t) \rangle \tag{7}
$$

*Proof.* The function $\mathbb{V}_e^t(Q)$ is convex with respect to $Q$, hence for any $Q_i^t$ and $Q_i^\star$ we have

$$
\mathbb{V}_e^t(Q_i^t) - \mathbb{V}_e^t(Q_i^\star) \leq \langle Q_i^t - Q_i^\star, \nabla_{Q_i^t} \mathbb{V}_e^t(Q_i^t) \rangle. \tag{8}
$$

Multiplying both sides of this inequality by $1 - \eta$, we have

$$
(1 - \eta)\mathbb{V}_e^t(Q_i^t) - (1 - \eta)\mathbb{V}_e^t(Q_i^\star) \tag{9}
$$
$$
\leq \langle Q_i^t - Q_i^\star, \nabla_{Q_i^t} \mathbb{V}_e^t(Q_i^t) \rangle - \eta \langle Q_i^t - Q_i^\star, \nabla_{Q_i^t} \mathbb{V}_e^t(Q_i^t) \rangle. \tag{10}
$$

In the following, we prove this Lemma in our two bandit settings: *adversary MAB setting* and *adversary MAB with multiple plays setting*.

In *adversary MAB setting*, we have

$$
\langle Q_i^t, \nabla_{Q_i^t} \mathbb{V}_e^t(Q_i^t) \rangle = - \sum_{j \in \mathcal{N}_i} q_{ij}(t) \frac{\alpha_{ij}^2}{k \cdot q_{ij}(t)^2} \|h_j(t)\|^2 \tag{11}
$$
$$
= -\mathbb{V}_e^t(Q_i^t) \tag{12}
$$

In *adversary MAB with multiple plays setting*, we use the approximated effective variance $\sum_{j_s \in \mathcal{N}_i} \frac{\alpha_{ij_s}}{q_{ij_s}} \|h_{j_s}\|^2$ derived in Proposition 2. For notational simplicity, we denote the approximated effective variance as $\mathbb{V}_e$ in the following. We have

$$
\langle Q_i^t, \nabla_{Q_i^t} \mathbb{V}_e^t(Q_i^t) \rangle = - \sum_{S_i \subset \mathcal{N}_i} Q_{i,S_i}^t \sum_{j_s \in S_i} \frac{\alpha_{ij_s}}{q_{ij_s}(t)^2} \|h_{j_s}\|^2 \tag{13}
$$
$$
= - \sum_{j_s \in \mathcal{N}_i} \frac{\alpha_{ij_s}}{q_{ij_s}(t)^2} \|h_{j_s}\|^2 \sum_{S_i : j_s \in S_i} Q_{i,S_i}^t \tag{14}
$$
$$
= - \sum_{j_s \in \mathcal{N}_i} \frac{\alpha_{ij_s}}{q_{ij_s}(t)} \|h_{j_s}\|^2 \tag{15}
$$
$$
= -\mathbb{V}_e^t(Q_i^t). \tag{16}
$$

The equation (13) holds because of Proposition 3.

At last, we conclude the proof

$$(1 - \eta)\mathbb{V}_e^t(Q_i^t) - (1 - \eta)\mathbb{V}_e^t(Q_i^\star) \tag{17}$$

$$\leq \langle Q_i^t - Q_i^\star, \nabla_{Q_i^t}\mathbb{V}_e^t(Q_i^t)\rangle - \eta\langle Q_i^t - Q_i^\star, \nabla_{Q_i^t}\mathbb{V}_e^t(Q_i^t)\rangle \tag{18}$$

$$= \langle Q_i^t - Q_i^\star, \nabla_{Q_i^t}\mathbb{V}_e^t(Q_i^t)\rangle + \eta\langle Q_i^\star, \nabla_{Q_i^t}\mathbb{V}_e^t(Q_i^t)\rangle + \eta\mathbb{V}_e^t(Q_i^t). \tag{19}$$

$\square$

**Theorem 1.** *Using Algorithm 1 with $\eta = 0.4$ and $\delta = \sqrt{(1 - \eta)\eta^4 k^5 \ln(n/k)/(Tn^4)}$ to minimize effective variance with respect to $\{Q_i^t\}_{1 \leq t \leq T}$, we have*

$$\sum_{t=1}^T \mathbb{V}_e^t(Q_i^t) \leq 3\sum_{t=1}^T \mathbb{V}_e^t(Q_i^\star) + 10\sqrt{\frac{Tn^4 \ln(n/k)}{k^3}} \tag{20}$$

*where $T \geq \ln(n/k)n^2(1 - \eta)/(k\eta^2)$ and $n = |\mathcal{N}_i|$.*

*Proof.* Without loss of generality, we prove the result by following the adversary MAB setting with DepRound and EXP3.M. First we explain why condition $T \geq \ln(n/k)n^2(1 - \eta)/(k\eta^2)$ ensures that $\delta\hat{r}_{ij}(t) \leq 1$,

$$\delta\hat{r}_{ij}(t) = \sqrt{\frac{(1 - \eta)\eta^4 k^5 \ln(n/k)}{Tn^4}} \cdot \frac{\alpha_{ij}(t)}{q_{ij}^3(t)}\|h_j(t)\|^2 \tag{21}$$

$$\leq \sqrt{\frac{(1 - \eta)\eta^4 k^5 \ln(n/k)}{Tn^4}} \cdot \frac{n^3}{k^3\eta^3} \tag{22}$$

$$\leq 1 \tag{23}$$

Assuming $\|h_j(t)\| \leq 1$, inequality (22) holds because $\alpha_{ij}(t) \leq 1$ and $q_{ij}(t) \geq k\eta/n$. Then replace $T$ by the condition, we get $\delta\hat{r}_{ij}(t) \leq 1$.

Let $W_i(t)$, $W_i'(t)$ denote $\sum_{j \in \mathcal{N}_i} w_{ij}(t)$, $\sum_{j \in \mathcal{N}_i} w_{ij}'(t)$ respectively. Then for any $t = 1, 2, ..., T$,

$$\frac{W_i(t+1)}{W_i(t)} = \sum_{j \in \mathcal{N}_i \backslash U_i^t} \frac{w_{ij}(t+1)}{W_i(t)} + \sum_{j \in U_i^t} \frac{w_{ij}(t+1)}{W_i(t)} \tag{24}$$

$$= \sum_{j \in \mathcal{N}_i \backslash U_i^t} \frac{w_{ij}(t)}{W_i(t)} \cdot \exp(\delta\hat{r}_{ij}(t)) + \sum_{j \in U_i^t} \frac{w_{ij}(t)}{W_i(t)} \tag{25}$$

$$\leq \sum_{j \in \mathcal{N}_i \backslash U_i^t} \frac{w_{ij}(t)}{W_i(t)}\left[1 + \delta\hat{r}_{ij}(t) + (\delta\hat{r}_{ij}(t))^2\right] + \sum_{j \in U_i^t} \frac{w_{ij}(t)}{W_i(t)} \tag{26}$$

$$= 1 + \frac{W_i'(t)}{W_i(t)} \sum_{j \in \mathcal{N}_i \backslash U_i^t} \frac{w_{ij}(t)}{W_i'(t)}\left[\delta\hat{r}_{ij}(t) + (\delta\hat{r}_{ij}(t))^2\right] \tag{27}$$

$$= 1 + \frac{W_i'(t)}{W_i(t)} \sum_{j \in \mathcal{N}_i \backslash U_i^t} \frac{q_{ij}(t)/k - \eta/n}{1 - \eta}\left[\delta\hat{r}_{ij}(t) + (\delta\hat{r}_{ij}(t))^2\right] \tag{28}$$

$$\leq 1 + \frac{\delta}{k(1 - \eta)} \sum_{j \in \mathcal{N}_i \backslash U_i^t} q_{ij}(t)\hat{r}_{ij}(t) + \frac{\delta^2}{k(1 - \eta)} \sum_{j \in \mathcal{N}_i \backslash U_i^t} q_{ij}(t)\hat{r}_{ij}^2(t) \tag{29}$$

Inequality (26) uses $e^a \leq 1 + a + a^2$ for $a \leq 1$. Equality (28) holds because of update equation of $q_{ij}(t)$ defined in EXP3.M. Inequality (29) holds because $\frac{W_i'(t)}{W_i(t)} \leq 1$. Since $1 + x \leq e^x$ for $x \geq 0$, we have

$$\ln\frac{W_i(t+1)}{W_i(t)} \leq \frac{\delta}{k(1 - \eta)} \sum_{j \in \mathcal{N}_i \backslash U_i^t} q_{ij}(t)\hat{r}_{ij}(t) + \frac{\delta^2}{k(1 - \eta)} \sum_{j \in \mathcal{N}_i \backslash U_i^t} q_{ij}(t)\hat{r}_{ij}^2(t) \tag{30}$$

If we sum, for $1 \leq t \leq T$, we get the following telescopic sum

$$\ln \frac{W_i(T+1)}{W_i(1)} = \sum_{t=1}^{T} \ln \frac{W_i(t+1)}{W_i(t)} \tag{31}$$

$$\leq \frac{\delta}{k(1-\eta)} \sum_{t=1}^{T} \sum_{j \in \mathcal{N}_i \setminus U_i^t} q_{ij}(t)\hat{r}_{ij}(t) + \frac{\delta^2}{k(1-\eta)} \sum_{t=1}^{T} \sum_{j \in \mathcal{N}_i \setminus U_i^t} q_{ij}(t)\hat{r}_{ij}^2(t) \tag{32}$$

$$\leq \frac{\delta}{k(1-\eta)} \sum_{t=1}^{T} \sum_{j \in \mathcal{N}_i \setminus U_i^t} q_{ij}(t)\hat{r}_{ij}(t) + \frac{\delta^2}{k(1-\eta)} \sum_{t=1}^{T} \sum_{j \in \mathcal{N}_i} q_{ij}(t)\hat{r}_{ij}^2(t) \tag{33}$$

On the other hand, for any subset $S$ containing k elements,

$$\ln \frac{W_i(T+1)}{W_i(1)} \geq \ln \frac{\sum_{j \in S} w_{ij}(T+1)}{W_i(1)} \tag{34}$$

$$\geq \frac{\sum_{j \in S} \ln w_{ij}(T+1)}{k} - \ln \frac{n}{k} \tag{35}$$

$$\geq \frac{\delta}{k} \sum_{j \in S} \sum_{t:j \notin U_i^t} \hat{r}_{ij}(t) - \ln \frac{n}{k} \tag{36}$$

The inequality (35) uses the fact that

$$\sum_{j \in S} w_{ij}(T+1) \geq k \left(\prod_{j \in S} w_{ij}(T+1)\right)^{1/k}$$

The equation (36) uses the fact that

$$w_{ij}(T+1) = \exp\left(\delta \sum_{t:j \notin U_i^t} \hat{r}_{ij}(t)\right)$$

From (33) and (36), we get

$$\frac{\delta}{k} \sum_{j \in S} \sum_{t:j \notin U_i^t} \hat{r}_{ij}(t) - \ln \frac{n}{k} \leq \frac{\delta}{k(1-\eta)} \sum_{t=1}^{T} \sum_{j \in \mathcal{N}_i \setminus U_i^t} q_{ij}(t)\hat{r}_{ij}(t) + \frac{\delta^2}{k(1-\eta)} \sum_{t=1}^{T} \sum_{j \in \mathcal{N}_i \setminus U_i^t} q_{ij}(t)\hat{r}_{ij}^2(t) \tag{37}$$

And we have the following inequality

$$\frac{\delta}{k} \sum_{j \in S} \sum_{t:j \in U_i^t} r_{ij}(t) = \frac{\delta}{k} \sum_{j \in S} \sum_{t:j \in U_i^t} q_{ij}(t)\hat{r}_{ij}(t) \tag{38}$$

$$\leq \frac{\delta}{k(1-\eta)} \sum_{t=1}^{T} \sum_{j \in U_i^t} q_{ij}(t)\hat{r}_{ij}(t) \tag{39}$$

The equality (38) holds beacuse $r_{ij}(t) = q_{ij}\hat{r}_{ij}(t)$ when $j \in S_i^t$ and $U_i^t \subseteq S_i^t$ bacause $q_{ij}^t = 1$ for all $j \in U_i^t$.

Then add inequality (39) in (37) we have

$$\frac{\delta}{k} \sum_{j \in S} \sum_{t:j \in U_i^t} r_{ij}(t) + \frac{\delta}{k} \sum_{j \in S} \sum_{t:j \notin U_i^t} \hat{r}_{ij}(t) - \ln \frac{n}{k} \tag{40}$$

$$\leq \frac{\delta}{k(1-\eta)} \sum_{t=1}^{T} \sum_{j \in \mathcal{N}_i} q_{ij}(t)\hat{r}_{ij}(t) + \frac{\delta^2}{k(1-\eta)} \sum_{t=1}^{T} \sum_{j \in \mathcal{N}_i} q_{ij}(t)\hat{r}_{ij}^2(t) \tag{41}$$

Given $q_{ij}(t)$ we have $\mathbb{E}[\hat{r}_{ij}^2(t)] = r_{ij}^2(t)/q_{ij}(t)$, hence, taking expectation of (40) yields that

$$\frac{\delta}{k} \sum_{t=1}^{T} \sum_{j \in S} r_{ij}(t) - \ln \frac{n}{k} \leq \frac{\delta}{k(1-\eta)} \sum_{t=1}^{T} \sum_{j \in \mathcal{N}_i} q_{ij}(t) r_{ij}(t) + \frac{\delta^2}{k(1-\eta)} \sum_{t=1}^{T} \sum_{j \in \mathcal{N}_i} r_{ij}^2(t) \tag{42}$$

By multiplying (42) by $Q_{i,S}^\star$ and summing over $S$, we get

$$\frac{\delta}{k}\sum_{t=1}^T\sum_{S\subset\mathcal{N}_i}Q_{i,S}^\star\sum_{j\in S}r_{ij}(t) - \ln\frac{n}{k} \le \frac{\delta}{k(1-\eta)}\sum_{t=1}^T\sum_{j\in\mathcal{N}_i}q_{ij}(t)r_{ij}(t) + \frac{\delta^2}{k(1-\eta)}\sum_{t=1}^T\sum_{j\in\mathcal{N}_i}r_{ij}^2(t) \quad (43)$$

As

$$\sum_{j\in\mathcal{N}_i}q_{ij}(t)r_{ij}(t) = \sum_{j\in\mathcal{N}_i}\sum_{S_i:j\in S_i}Q_{i,S_i}^t r_{ij}(t) \tag{44}$$

$$= \sum_{S_i\subset\mathcal{N}_i}Q_{i,S_i}^t\sum_{j\in S_i}r_{ij}(t) \tag{45}$$

$$= -\sum_{S_i\subset\mathcal{N}_i}Q_{i,S_i}^t\nabla_{Q_{i,S_i}^t}\mathbb{V}_e^t(Q_{i,S_i}^t) \tag{46}$$

$$= -\langle Q_i^t, \nabla_{Q_i^t}\mathbb{V}_e^t(Q_i^t)\rangle \tag{47}$$

By plugging (47) in (43) and rearranging it, we find

$$\sum_{t=1}^T\langle Q_i^t - Q_i^\star, \nabla_{Q_i^t}\mathbb{V}_e^t(Q_i^t)\rangle + \eta\sum_{t=1}^T\langle Q_i^\star, \nabla_{Q_i^t}\mathbb{V}_e^t(Q_i^t)\rangle \tag{48}$$

$$\le \delta\sum_{t=1}^T\sum_{j\in\mathcal{N}_i}r_{ij}^2(t) + \frac{(1-\eta)k}{\delta}\ln(n/k)$$

Using Lemma 1, we have

$$(1-2\eta)\sum_{t=1}^T\mathbb{V}_e^t(Q_i^t) - (1-\eta)\sum_{t=1}^T\mathbb{V}_e^t(Q_i^\star) \le \delta\sum_{t=1}^T\sum_{j\in\mathcal{N}_i}r_{ij}^2(t) + \frac{(1-\eta)k}{\delta}\ln(n/k) \tag{49}$$

Finally, we know that

$$\sum_{j\in\mathcal{N}_i}r_{ij}^2(t) = \sum_{j\in\mathcal{N}_i}\frac{\alpha_{ij}(t)^2}{q_{ij}(t)^4} \tag{50}$$

$$\le \sum_{j\in\mathcal{N}_i}\alpha_{ij}(t)\frac{n^4}{k^4\eta^4} \ (because \ q_{ij}(t)\ge k\eta/n) \tag{51}$$

$$= \frac{n^4}{k^4\eta^4} \tag{52}$$

By setting $\eta = 0.4$ and $\delta = \sqrt{(1-\eta)\eta^4 k^5 \ln(n/k)/(Tn^4)}$, we get the upper bound.

$\square$