[Reviews · NeurIPS 2020]

Review 1

Summary and Contributions: In this paper, the authors studied the sampling and its variance for GCNs and GNNs. They proposed to formulate the optimization of the sampling variance as an adversary bandit problem. Experimental results on several benchmark datasets demonstrated the effectiveness of the proposed method. ========================= post-rebuttal edit ========================= Thank the author's response much. It addressed part of my concerns. I'd like to keep my score. My main concerns are still related to the experiments. I encourage the authors to try more baselines and datasets to further verify the advantages of the proposed method.

Strengths: 1. The research problem of studying how to reduce the sampling variance of graph-structured data is interesting. 2. This paper seems theoretically solid, with very rich and detailed analysis. 3. Experimental results on benchmark datasets indicate the effectiveness of the proposed method.

Weaknesses: 1. Although the problem studied in this paper is interesting, this pape is not easy to follow. 2. More baselines (sampling approaches designed for GNNs) are needed. In Table 2, S-GCN [5] is a simple sampler. ClusterGCN and GraphSAINT are designed for sampling (sub)graphs. The same for Table 3. 3. To be honest, I am kind of confused about Table 3. It would be better if the authors provide more analysis for Table 3. And more analysis when Tables 2,3 are considered together. 4. How did the authors determine the hyper-parameter settings? 5. I am interested in seeing more experimental comparison on the datasets with a large number of nodes. 6. How about the comparison in terms of computation cost / running time?

Correctness: It seems to be correct.

Clarity: Yes, but it is not easy to follow.

Relation to Prior Work: Yes.

Reproducibility: No

Additional Feedback:


Review 2

Summary and Contributions: The authors propose to use a bandit approach to optimally sample the neighbors in GNN embeddings. Previous approaches include random and importance sampling and proposed approach scales even to GNNS with attention since they can change across iterations. Also a nice theoretical bound shows a multiplicative factor of 3 over the optimal variance.

Strengths: (A) Good theoretical grounding in bandits using multi-play MAB is proposed for neighbor selection and a variance bound is shown. (B) Strong empirical results as shown on a wide-variety of datasets. (C) Casting node selection to a bandit problem seems novel to me and could lead to other bandit extensions as well as applications to other graph settings. (D) Highly relevant since training GNNs is expensive and improvements are highly welcome.

Weaknesses: (A) How about using combinatorial bandits (CMAB) for selecting the neighbors? (B) I might have missed it but further insight into why this approach works could have been interesting. Are you able to better sample neighbors in fewer rounds while random/importance sampling explore unnecessarily?

Correctness: I haven't checked all the proofs but its seems to be correct.

Clarity: Yes, the paper is clearly laid out and easy to follow. Could use an additional pass though.

Relation to Prior Work: The prior work is discussed and also extensively compared in the experiments.

Reproducibility: Yes

Additional Feedback: Update after discussion: The authors answered most of the questions and there seems to be some concern about novelty. However, keeping my score since its application to GNN is interesting and the fact both attention weights and embeddings change and are handled in the paper.


Review 3

Summary and Contributions: This paper studies the problem of accelerating the training of graph neural networks (GNNs) using sampling. The embedding of a node is computed by aggregating the embeddings of its neighboring nodes. Instead of sampling all neighbors - which can be expensive - we can sample a subset of the nodes. The resulting estimator's variance can be reduced using importance sampling. For GNNs, it is difficult to determine the optimal importance sampling weights since they depend on unknown quantities. This paper proposes the use of adversarial bandits methods to learn the optimal sampling weights.

Strengths: The paper is relevant to the neurips community. The paper is well-written, and the authors have compared their sampling scheme to existing sampling schemes. They also provide a regret guarantee for their algorithm. The contribution is novel to the best of my knowledge.

Weaknesses: I do not follow why the gradient of the variance can be set as the reward for the bandit (the writeup after equation 7). The objective is to choose a good Q_i^t, and the quantity on the lhs of equation 7 is to be minimized. This is upper bounded by the inner product on the rhs. In the reward function however, only the gradient is present. Could you elaborate this? In the experiments, I would have liked to see two plots. First, a plot comparing the empirical regret with the upper bound in Theorem 1. This would help validate the expression in Theorem 1 and understand whether it predicts the dependence on various quantities correctly. Second, a plot measuring the gains on simulated toy graphs. For instance, how does the convergence change if we consider graphs where all nodes have a fixed degree. I would imagine that as the degree increases, the convergence would be slower (for a fixed sample of the neighborhood) I would also encourage releasing the code for reproducibility.

Correctness: Yes. As long as the authors can answer the question about the gradient being chosen as the reward above.

Clarity: Yes

Relation to Prior Work: Yes. For completeness, the authors may look at the work by David Tse's group on using bandits as samplers (although they use it in the stochastic setting, not the adversarial setting).

Reproducibility: Yes

Additional Feedback: == after rebuttal == Thank you for clarifying eqn 7 and the plots.


Review 4

Summary and Contributions: The paper describes about the variance reduced samplers for training GCNs and attentive GNNs by formulating the optimisation of samplers as a bandit problem and proposes two multi-armed bandit algorithms.

Strengths: The paper is written well. The idea of using mutli-arm bandit for sampling the neighbours training of GCNs and attentive GNNs is interesting. The technical details and equations appears to be correct. The authors applied their multi- arm bandit approach to GAT and GeniePath and shown its effective over layer sampling approaches.

Weaknesses: The authors conduct experiments with 2 layer architecture. However, for few problems and dataset, it may require more complex architecture and the authors could clarify how the proposed algorithm performs in terms of scalability and computation cost. The notation used in the paper sometimes can be confusing. For example - in equation 4 - alpha_ij is a value not a function - alpha_ij(t) can be noted as function of 't' It is mentioned that the rewards vary as the training proceeds and it would have been interesting to explore how any of simple bandit algorithms perform in the experiments or how to apply simple bandit algorithms for the current experiments. The authors could try and adapt a simple eps-greedy method to solve this problem The algorithm based on MAB where a combinatorial set of neighbors with size k needs to be chosen. The size parameters 'k' will be a hyperparameter for the model which needs to tuned using validation set. The authors can provide clarification on how to select the size k.

Correctness: The technical content of paper appears to be correct. There could be additional experiments conducted to see how complex architecture can benefit such sampling approach for effective training of the model

Clarity: The paper is written and some notation in the equations can be clarified.

Relation to Prior Work: The authors summarize three types of related works for training graph neural networks. The authors clearly mentioned and highlighted their contributions with respect to related work.

Reproducibility: No

Additional Feedback:

[Author Response · NeurIPS 2020]

**R #1 and #5**: "computation cost". Our bandit sampler only requires additionally $E$ floats to store the alternative sampling distribution that will be optimized, where $E$ is the number of edges used for the message passing operations in GNN. Beyond that, no further storage is required. This is true even for more sophisticated architectures where messages are passed between neighbors beyond one hop. The bandit sampler has the same time complexity as any other node-wise sampling approaches. Fig. 1 (left) shows the runtime on PPI using deeper GCNs.

**R #2 and #5**: "combinatorial bandits" or "how other simple bandit algorithm such as eps-greedy can benefit, and how to apply". The major contribution of this work focuses on the formulation of neighbor selection as a bandit problem. We believe using other more sophisticated combinatorial bandits should be promising, and is left as future study. We can simply adapt epsilon greedy algorithm to our bandit framework. We have alternative sampling probability for the arm with the largest reward as $\frac{\epsilon}{K} + 1 - \epsilon$, and each of rest $(K - 1)$ arms as $\frac{\epsilon}{K}$. We can estimate the reward of arm $j$ by $\frac{\text{sum of rewards when arm } j \text{ taken before } t}{\text{number of times when arm } j \text{ taken before } t}$ to determine the arm with largest reward before sampling. We setup a simple test. We have a list of numbers [1,2,3,4,5,6,7,8] with mean $4.5$. We aim to estimate the mean by sampling with sample size 2. After 1000 times of sampling, we hope to have unbiased estimate with low variance. We compare three sampling approaches, i.e. random, EXP3 and eps-greedy. All of them are unbiased estimators, however, the variance of EXP3 is 0.25, while the variances of eps-greedy and random are 0.63 and 1.22 respectively. Finally, We show the convergence of our bandit approach with epsilon-greedy and EXP3 on PPI in Fig. 1 (middle).

**R #1 and #5**: "hyperparameter", "how to select sample size k". We perform grid search or follow previous literature for setting the hyparameters. We elaborate the hyperparameter setting in the supplement. We note that Theorem 1 shows that our algorithm can converge to optima using any sample size $k$. We simply select a small sample size $k$ to report the numbers, and do the sample size analysis in section 7.3 as well.

**R #1**: **(1)** "More baseline approaches are needed." To the best of our knowledge, we have tried our best to include all state-of-the-art approaches in the related works and experiments. **(2)** "Table 3 is confused." The comparison is fair only if we compare the proposed sampling approach with other sampling approaches under the *identical* neural architectures. So we report the results of GCNs and attentive GNNs in table 2 and 3 respectively. There are no sampling approaches work for attentive GNNs that can guarantee sampling variance and unbiased estimates to our best knowledge. We promise to include more analyses on those numbers. **(3)** "more experiments on large dataset". We have tested our approach on several large OGBN datasets. For example, our bandit sampler obtained 0.7825 on OGBN-proteins (132,534 nodes) dataset and 0.803 on OGBN-products (2,449,029 nodes) dataset, which significantly outperform other sampling approaches using the same GNN architecture. We will include these numbers in the paper. **(4)** "running time". We have reported the convergence in terms of running time in the supplementary material.

**R #2**: **(1)** "insight into why bandit works". Optimize the variance help the convergence. However, optimal variance requires the knowledge of all the neighbors' embeddings that are computation infeasible. Our chance is to exploit the sampled good neighbors. This motivates us using bandit algorithms.

**R #3**: **(1)** "why set gradient of the variance as the reward". The variance difference (lhs on eq.7) to be minimized is upper bounded by the rhs due to convex property. The rhs on eq.7 can be interpreted as expected loss (negative reward) with policy $Q_i^t$ minus expected loss with optimal policy. Hence, the negative derivative of variance is reward in case we optimize this rhs. Moreover, this upper bound yields lemma 1 in the supplement and lets us prove the result in theorem 1. **(2)** "plots on toy data to show the convergence as degree increases". We set up the following experiments. We randomly sample 100 labeled nodes $\{1,...,i,...,100\}$ with each $\mu_i$ uniformly sampled from [-10, 10]. For each labeled node $i$ we generate $k$ neighbors, and its neighbors' features are 1-dimensional scalars in real field that are sampled from $\text{uniform}(\mu_i - \sigma, \mu_i + \sigma)$. Each node $i$'s label is generated by simply averaging its neighbors' 1-dimensional scalar features. We compare the convergence with a random sampler by increasing $k = 50$ to 100 and 200 in Fig. 1 (right). All samplers use a fixed sample size 10. **(3)** "a plot comparing the empirical regret with the upper bound". The optimal variance is hard to compute, but can be viewed as a constant. We will show the variances w.r.t epochs in revision.

Figure 1: **left**: deeper arch running time; **middle**: EXP3 vs. eps-greedy; **right**: convergence on toy data.

[Meta-Review · NeurIPS 2020]

There are several papers dealing with the issue of sampling when training a graph neural network, and this paper provides a new approach to it using (adversarial) bandit technology. This idea makes sense and the authors provide convincing arguments for why this technique should improve upon previous art. Although there may be room for slightly more comprehensive experiments and reasoning for the details of the proposed solution, the paper provides a clear representation of a novel method that works well for an important problem. This paper would make a good addition to NeurIPS.